# Biomarkers of Cancer Stem Cells for Experimental Research and Clinical Application

**DOI:** 10.3390/jpm12050715

**Published:** 2022-04-29

**Authors:** Shigeo Saito, Chia-Chen Ku, Kenly Wuputra, Jia-Bin Pan, Chang-Shen Lin, Ying-Chu Lin, Deng-Chyang Wu, Kazunari K. Yokoyama

**Affiliations:** 1Saito Laboratory of Cell Technology, Yaita 329-1571, Japan; 2Horus Co., Ltd., Nakano, Tokyo 164-0001, Japan; 3Graduate Institute of Medicine, Department of Medicine, Kaohsiung Medical University, Kaohsiung 80708, Taiwan; r991046@gap.kmu.edu.tw (C.-C.K.); kenlywu@hotmail.com (K.W.); r060139@gap.kmu.edu.tw (J.-B.P.); changshen.lin@gmail.com (C.-S.L.); 4Regenerative Medicine and Cell Therapy Research Center, Kaohsiung Medical University, Kaohsiung 80708, Taiwan; 5Cell Therapy and Research Center, Kaohsiung Medical University Hospital, Kaohsiung Medical University, Kaohsiung 80756, Taiwan; 6School of Dentistry, Department of Dentistry, Kaohsiung Medical University, Kaohsiung 80708, Taiwan; culin@cc.kmu.edu.tw; 7Division of Gastroenterology, Department of Internal Medicine, Kaohsiung Medical University Hospital, Kaohsiung Medical University, Kaohsiung 80756, Taiwan

**Keywords:** biomarkers, cancer progression, cancer stem cells, cell plasticity, microenvironment, reprogramming factors, stem cell markers

## Abstract

The use of biomarkers in cancer diagnosis, therapy, and prognosis has been highly effective over several decades. Studies of biomarkers in cancer patients pre- and post-treatment and during cancer progression have helped identify cancer stem cells (CSCs) and their related microenvironments. These analyses are critical for the therapeutic application of drugs and the efficient targeting and prevention of cancer progression, as well as the investigation of the mechanism of the cancer development. Biomarkers that characterize CSCs have thus been identified and correlated to diagnosis, therapy, and prognosis. However, CSCs demonstrate elevated levels of plasticity, which alters their functional phenotype and appearance by interacting with their microenvironments, in response to chemotherapy and radiotherapeutics. In turn, these changes induce different metabolic adaptations of CSCs. This article provides a review of the most frequently used CSCs and stem cell markers.

## 1. Introduction

It is believed that uncontrolled progression of tumor cells is generated by a small population of cancer stem cells (CSCs) that possess the capability for self-renewal and pluripotent differentiation into multiple cancer cell types [1]. CSCs are hypothesized to persist in cancers and cause metastasis, therapy resistance, and post-operative recurrence by producing new tumor cells. CSCs can survive many commonly employed treatments [2]. Accordingly, targeting CSCs should provide new therapies to improve survival of cancer patients. Moreover, this research field may identify the heterogeneity of tumor cell populations and their genetic, epigenetic, and microenvironmental diversification.

The existence of CSCs was first identified in acute myeloid leukemia in 1997 [3], although the terminology was first employed by Reya and colleagues in 2001 [4]. CSCs were then found in glioblastoma [5], breast carcinomas [6,7], gastric cancer [8], and colorectal cancer [9]. However, until now, the mechanisms that drive the intracellular dysregulation of CSCs to malignancy have remained unclear. Biomarkers of these CSCs are thus critical for investigating the mechanisms by which CSCs can develop into neoplasia interacting with the surrounding cells. In addition, these markers are useful for both identifying the heterogeneity of CSCs and determining their cell fates.

## 2. Common Features of Cancer Formation and the Cell Reprogramming Process in Induced Pluripotent Stem Cells (iPSCs)

To establish induced pluripotent stem cells (iPSCs) from somatic cells, overexpression of pluripotency-related transcription factors—such as octamer-binding transcription factor 4 (OCT4), Krüppel-like factor 4 (KLF4), sex determining region Y-box 2 (SOX2), cellular myelocytomatosis virus gene (c-MYC) (OKSM), homeobox nanog transcription factor (NANOG), and lin-28 homolog (LIN28)—is a necessary part of the methodology for reprogramming differentiated cells to iPSCs [10,11,12]. The risk of tumor formation is considered in the process of cell reprogramming and can lead to tumorigenesis, due to cells acquiring the capabilities of self-renewal and de-differentiation similarly to stem cells [13]. Cancer cells generally arise from normal cells that have undergone severe alterations at the genetic, epigenetic, and microenvironmental levels [14]. A lot of cancer cells can be produced by a series of mutations in their DNA sequences that induce uncontrolled cell proliferation. Therefore, cancer can be defined as a dysfunction of the organization of cellular and tissue development in an individual’s body. The cell reprogramming process seems to share common features with cancer formation, including indefinite cell proliferation and self-renewal capability. These similar features indicate that the process of reprogramming and cancer formation utilize overlapping molecular signaling and epigenetic pathways [15,16]. Other reports suggest that cancer progression is caused by the occurrence of tumorigenic enhancer-reactivation in both somatic cells and cancer cells [17,18]. An important pluripotency-inducing transcription factor, Oct4, is necessary to maintain the characteristics of murine embryonic stem cells (ESCs), and Oct4 knockout mice cannot generate an inner cell mass, and thus differentiate into trophectoderm [19]. However, a high level of OCT4 expression leads to a poor prognosis in various types of cancer, such as bladder cancers [20]; cancers of the ovaries, pancreas, and testicles [21]; medulloblastoma [22]; and esophageal squamous cell carcinoma [23]. Other cell reprogramming factors are also detected in many types of cancers. For instance, KLF4 is known as a prognostic predictor of colon cancer [24], and is expressed in leukemia, testicular cancer [21], and breast cancer [25]. Other reports suggest an adverse function of KLF4, as loss of cytoplasmic or nuclear expression of KLF4 was related to a poor prognosis in nasopharyngeal carcinoma and oral cancer [26,27]. High expression of SOX2 was associated with poor prognosis in esophageal squamous cell carcinoma [28], gastric carcinoma [29,30], breast cancer [31], and testicular cancer [32]. In addition, c-MYC overexpression is associated with human cancers, including breast cancer, colon cancer, glioma, medulloblastoma, pancreatic cancer, prostate cancer, and hepatocellular carcinoma [21,33,34]. Expression of pluripotency-regulating transcription factors—such as OCT4, SOX2, and NANOG (OSN)—in patients predicted poor clinical outcomes and resistance to care treatment [35]. These findings indicate that pluripotency-inducing transcription factors can be evaluated as proto-oncogenes and biomarkers of cancers [36,37], but the mechanisms by which these factors provoke cancer initiation and malignant cancer progression are poorly understood. However, it is likely that a loss of control of genes that regulate the cell cycle and a series of mutations in proto-oncogenes and tumor suppressor genes that produce uncontrolled cell division are required for the induction of transformation to cancer cells [38].

## 3. Cancer Cell Reprogramming for Cancer Initiation Modeling

Shortly after the establishment of human somatic cell-derived iPSCs, a similar reprogramming method was adapted to various types of cancer cells. This technology is useful for verifying molecular and epigenetic alterations, as well as sensitivity to drug resistance, between cancers and stemness to gain a better understanding of the acquired CSCs. There are several reports demonstrating that induced pluripotent cancer cells (iPCCs) or induced pluripotent cancer stem cells (iPCSCs) display a more serious cancer phenotype because of the activity of oncogenic reprogramming factors. With OKSM reprogramming factors, iPCCs derived from chronic leukemia KBM-7 cells exhibited resistance to an inhibitor of oncogenes in these cells, but not in parental cancer cells [39]. The iPCCs established from patients who suffered from chronic myelogenous leukemia (CML) cancer and were sensitive to the anti-cancer kinase inhibitor, imatinib, finally exhibited drug resistance, displaying a resemblance to CML stem cells [40]. These results thus present a model system of drug resistance and the characteristics of CSCs. Similarly, for gastrointestinal cancer cell lines that underwent reprogramming using OKSM, the derived iPCCs had more aggressive features than the parental cells with increased time in culture [39]. These authors presumed that the cancer-specific iPCCs were subjected to genetic instability via genetic or epigenetic alterations, including oncogenic activation of c-MYC.

In addition, we produced iPCCs from a DAOY medulloblastoma cell line using OCT4 and Jun dimerization protein 2 (JDP2) transcription factors for reprogramming. The derived iPCCs displayed a higher tumorigenic competence of xenografts than those from parental DAOY cells, and the CSC-like features of iPCCs produced via forced expression of JDP 2 were confirmed in this experiment [41]. Jdp2 is known to have the opposite action to c-Jun in mouse embryonic stem cells (mESC). In the case of HepG2 hepatoblastoma cells, the combination of OCT4 and c-JUN generated CSC-like cells from HepG2 cell lines, as reported [42]. These cell clones seem to be CSCs, as only 10 cells can induce tumor formation in SCID mice. Furthermore, iPCCs should be valuable in research on drug screening or cancer initiation mechanisms in the field of human cancer therapeutics.

Some reports have described the repression of tumorigenicity in reprogrammed iPCCs. Miyoshi et al. [43] generated gastrointestinal cancer cell (GCC)-derived iPSC-like cells via ectopic expression of OKSM and B-cell lymphoma 2 (*BCL2*) and Kirsten rat sarcoma virus (*kRAS*) oncogenes, together with short-hairpin RNAs (shRNAs) for antitumor suppressor genes such as tumor protein p53 (*TP53*), cyclin-dependent kinase inhibitor 2A (CDKN2A = *p16^ink4a^)*, phosphatase and tensin homolog deleted on chromosome 10 (*PTEN*), fragile histidine triad di adenosine triphosphatase (*FHIT*), and retinoblastoma protein 1 *(RB1*). The iPSC-like cells proved to be more sensitive to 5-fluorouracil and displayed decreased tumorigenicity in immunodeficient mice. This study indicates that the controversial method of using cancer-initiating genes in reprogramming could contribute to the development of a new therapy to eliminate residual CSCs. These differences in iPCCs between oncogenesis and anti-oncogenesis have not been characterized in detail. However, we speculate that the mutation of possible driver mutated genes might play a critical role in the commitment to cell fates. These genes might include *Apc*, *p53*, *Kras*, *Pten*, or *Smad*, which have been reported to generate invasive tumors originating from stem cells [44]. In addition, the signaling of Wnt/TCF, cadherins, STAT3, and NF-κB plays an important role in the control of cell proliferation, differentiation, death, senescence, and plasticity. Therefore, stemness factors in the steady state are critical for producing vast changes in the transcription and DNA damage/mismatch/repair machineries, including epigenesis, DNA methylation, mitochondrial DNA, and metabolism. These changes are recognized by surface biomarkers such as CD44, CD133, CD34, and the epithelial cell adhesion molecule (EpCAM) series to characterize the transition to CSCs.

## 4. Markers of Putative Cellular Targets for Therapeutics in CSCs

The risk of cancer initiation is dependent on mutations of oncogenes and anti-oncogenes during the conversion of normal stem cells into cancer cells and on the environmental effects of stem cells [1,45]. As in cases of general cancer cells, several studies have reported expression levels of pluripotent factors as markers for stemness in CSCs. It was demonstrated that overexpression of c-MYC in immortalized mammary epithelial cells favored the onset of tumorigenesis via epigenetic cell reprogramming [46], and that KLF4 acted as an oncogene in colon CSCs [47]. Breast CSCs were found to have a self-renewal ability and common features of gene expression with ESCs. Expression of OCT4, SOX2 [31], and NANOG [48] was detected in breast CSCs, in order for them to maintain their stem cell nature. Lu et al. [49] discovered that NANOG overexpression in breast CSCs increased the expression of stemness factors such as OCT4, KLF, and SOX2. The authors found that NANOG and hypoxia-inducible factor 1α (HIF1α) cooperated in the breast CSC specification triggered by hypoxia through the activation of the telomerase reverse transcriptase (TERT) gene. Murine leukemia virus insertion site 1 (Bmi1) was demonstrated to control self-renewal of CSCs and to function in human head and neck squamous cell carcinoma (HNSCC) [2]. Recently, genetic inhibition of BMI1 was found to eliminate BMI1-expressing CSCs, resulting in the prevention of metastatic tumor growth and relapse in HNSCC [50].

These data suggest that simultaneous inhibition of stemness marker genes—such as *OCT4*, *KLF4*, *SOX2*, *c-MYC*, and *NANOG*, as well as *TERT*, *JDP2, HIF1α*, and *BMI1*—could become an effective means of cancer treatment by depriving residual CSCs of their self-renewal capacity.

## 5. Cell Surface and Genetic Markers in CSCs

It has been argued that the expression of specific CSC markers aligns with features of CSCs such as chemoresistance and recurrence of invasive tumorigenicity [51,52]. Since CSC cell surface markers largely fail to distinguish normal tissue stem cells from CSCs in solid tumors, most cell surface markers are unsuitable as targets for antibody therapy [52]. For example, the transmembrane glycoprotein CD133 is recognized as a cell surface marker of neuroepithelial stem cells, but is also detected in CSCs in colorectal, lung, and liver cancers [53,54,55,56]. CD133 expression is not restricted to stem cells, and both CD133-positive and CD133-negative colon cancer cells are able to initiate tumors [57]. Thus, CD133 is not a specific marker of CSCs.

CD44 is a marker of normal fetal and adult hematopoietic stem cells [1] and has also been evaluated as a CSC marker. The expression of CD44 was confirmed in various types of cancers with stemness characteristics, including breast [6,58], prostate [50,59,60], colon [61,62], and pancreatic [63] cancer, and in head and neck squamous carcinomas [64]. Inhibition of CD44 prevented tumor progression from colorectal CSCs [65]. Furthermore, CD44+/CD24+ and CD44+/CD54+ cells were identified as markers of gastric CSCs [66]. CD326 (i.e., epithelial cell adhesion molecule; EpCAM) is expressed in epithelial tissues, germ and somatic stem cells, and cancer cells [67,68,69].

Leucine-rich repeat-containing G-protein coupled receptor 5 (LGR5) is expressed in somatic stem cells in intestine, colon, hair follicles, and ovaries [70]. LGR5 is also proposed to be a conclusive marker of colorectal CSCs in cases where it is co-expressed with CD44 and EpCAM [71]. The representative biomarkers specific for each cancer are summarized in Table 1.

How can we determine the possible biomarkers for CSCs during the initiation and progression stages of cancer development? Once the biomarkers have been identified and characterized, the exact status of these markers and the driver mutations—such as *Apc*, *p53*, *Kras*, *Pten*, or *Smad*—should be identified, because driver mutations can decide cell destiny, such as an oncogenic or anti-oncogenic status. Next-generation sequencing (NSG) technology has enabled the characterization of the genetic classification of cancers to define mutations as “drivers” or “passengers” [72] and identify the level of genomic instability. Driver mutations are defined as mutations that enhance cell proliferation and the growth of cancer cells, whereas passenger or hitchhiker mutations do not [73]. Therefore, to gain a better understanding of the usefulness of these biomarker genes for characterizing CSCs, the exact combination of mutations of driver genes and relevant biomarkers needs to be determined in detail. Moreover, the cells surrounding CSCs need to be targeted to assess whether the implementation of personalized therapy to eradicate senescent tumor cells, cancer-associated fibroblasts (CAFs), and tumor-associated microenvironments (TAMs), and other niches can protect against tumor recurrence.

## 6. Microenvironmental Factors and Interplay with CSC Signaling

The CSC niche is a part of the specialized tumor microenvironment and is an essential factor for maintaining the phenotypes of CSCs [45,74]. Cells within the CSC niche secrete factors that induce self-renewal of CSCs, stimulate angiogenesis, and recruit other cells that produce additional factors to drive tumor cell invasiveness and metastasis [74,75]. It is still difficult to target the microenvironment to inhibit growth and metastasis of CSCs because CSCs can arise within the niche, which affects their ability to evade the inherited immune response and survive. One model suggests that CSCs are produced from genetically and epigenetically altered stem cells or progenitor cells that reside in the niches and that they obtain tumorigenic progression properties for maintaining tumor mass.

CSCs might also be well adapted to niche microenvironments [76]. Stemness of CSCs is niche-dependent and may represent one of the phenotypic states obtainable by various cancer genotypes when they are supplied with specific environmental factors. Accumulating evidence has demonstrated that cancer invasion and metastasis is regulated by extracellular matrices (ECM) derived from the tumor microenvironment [77,78]. In gastric cancer, collagens are dysregulated in advanced stages, and these collagen genes are suggested to be useful prognostic markers to differentiate between mature malignant lesions and premalignant lesions [79,80]. The series of events occurring within the tumor niche includes increased levels of ECM remodeling enzymes; recruitment of cancer-associated fibroblasts (CAFs), immune cells, and other stroma cells; secretion of growth factors; induction of collagen depositions that lead to increased ECM stiffness; disorder of cell-to-cell adhesion; and upregulation of integrins. These all-cell events promote tumor metastasis and progression [81,82,83].

Interplay between cancer cells and endothelial and immune cells, mesenchymal stem cells (MSCs), and fibroblast-like stroma cells plays a major role in creating the complex microenvironment known as the tumor niche [84]. These cells can originate from cancer resident-stromal cells, and are able to transform into CAFs [85,86].

One study provided evidence that interleukin 6 (IL6) secreted by MSCs promoted increased expression of CD133 in CSCs of murine colorectal cancer via the JAK-STAT signaling pathway [87]. JAK-STAT signaling is important for regulating self-renewal of normal stem cells and preserving stem cells in their respective niches via control of various adhesion molecules [1,88].

IL6 and CXCL8 present MSCs culture-conditioned media induced expression of OCT4 and SOX2 in colorectal cancer cells and promoted tumor progression via adenosine monophosphate protein kinase (AMPK)-mediated NF-κB activation [89]. Mutual signaling between different pathways seems to influence self-renewal and tumor initiation. Intestinal cancer formation was shown to be driven by an interplay between NF-κB signaling and the WNT pathway. Elevated NF-κB signaling in intestinal epithelial cells enhanced WNT signaling and induced carcinogenic cell reprogramming that could produce CSC-like cells with cancer initiation capacity in murine models [8,84]. In a colorectal cancer model, IL6 and angiopoietin 1 secreted by MSCs provoked cancer cells to produce endothelin 1, resulting in the promotion of cancer angiogenesis [84]. However, most populations of human colorectal CSCs, under the hypoxic conditions within a xenograft after transplantation into mice, result in better survival following administration of chemotherapies [90].

Patient-derived tumor organoid models are useful for establishing systematic pre-clinical models of cancer heterogeneity for investigating the mechanisms of cancer development and anticancer drug resistance [89,90,91,92,93,94]. Cancer-derived organoid models may also provide the basic systems for investigating the cellular, molecular, and epigenetic regulation of the cancer microenvironment niche.

Among various tumor microenvironmental factors, the extracellular matrices have been proven to play a key role in metastasis and prognosis in gastric cancer. Therefore, further exploration of niche factors may be an effective strategy to identify novel markers for cancer initiation, development, invasion, and prognosis.

## 7. Perspective on the Therapeutic Use of Biomarkers

Based on existing data, the stream of the commitment of somatic cells and cancers cells to undergo reprogramming to produce normal stem cells and CSCs, as well as induced stem cells, can be summarized, to understand their stemness and their biomarkers (Figure 1).

In this review, we have focused on reprogramming factors as putative universal biomarkers of stem cells, because they play a role in stem cell pluripotency and their interactions with various co-factors can enable transcriptional versatility during development [95,96]. CSCs are known to be a small population with self-renewal capacity and differentiation potential that confers tumor relapse, metastasis, heterogeneity, multidrug resistance, and radiation resistance [97].

Stemness genes such as *OCT4*, *SOX2*, and *NANOG*; some signaling-related stemness proteins, including WNT, NF-κB, NOTCH, HEDGEHOC, JAK-STAT, PI3K/AKT/mTOR, TGF/SMAD, and PPAR; as well as cell communication microenvironments such as vascular niches, hypoxia, TAMs, CAFs, ECMs, and exosomes are critical to the regulation of CSCs. Drugs, small molecules, vaccines, antibodies, and chimeric antigen receptor-T (CAR-T) cells targeting these pathways have been generated to target CSCs [98]. How can stemness genes in normal cells be altered to cancer stemness genes using driver mutations? As previously described, the reprogramming factor, OCT4, is required for the generation of liver CSCs with the oncogene, *c-JUN* [23,42]. Moreover, the AP-1 repressor, JDP2, plays a dual role in the reprogramming of cancer cells [41]. Why do we observe the reversed role of JDP2? We found that the status of TP53 in cancer cells is critical for the generation of oncogenic and anti-oncogenic function [1,41,42,45]. This critical function of TP53 mutation in cancer cells was also reported to generate the reprogramming by other authors [43,44]. Expression of normal p53 in the original somatic cells is critical for normal reprogramming [99,100,101]. Therefore, the status of TP53 and/or the methylation status of *p16^Ink4a^* should be examined [1,44,45]. Stemness genes are mutated or not during the cellular reprogramming of cancer cells to generate CSCs. Therefore, the role of at least the *OCT4*, *SOX2*, and *NONOG* genes in the metabolic reprogramming, cell plasticity, and trans-differentiation processes should be clarified in future studies. Indeed, the functional role of stem cell factors in cancer commitment should be further investigated. Recently, the role of SOX2 has been reported [102].

For example, the introduction of SOX2 in prostate cancer induces stem-like characteristics but uses different metabolic pathways and interacts with different target gene products. The oncogenic role of SOX2 is confirmed further by several studies exhibiting SOX2 dependent alteration of cell growth, invasion ability, and chemo-resistant activity beyond tumor types [103,104]. de Wet et al. demonstrated that most target genes of SOX2 regulation in prostate cancer are non-overlapping with targets of SOX2 in human ESCs, and identified different *cis* elements within what appear to be similar target genes.

In addition, SOX2 employed the activities of canonical reprogramming, glycolytic, and oxidative phosphorylation reactions in prostate cancer cells different from those associated with canonical SOX2 function in ESCs. This suggests the relocation of SOX2 to novel oncogenic drivers during progression of prostate tumor [102]. Thus, stem cell markers are not universal, and are context dependent. Some reports pertained to the role of SOX2 in cancer metabolism [93]. Glucose metabolism in cancer exhibited a unique bioreaction whereby the main energy source of primary prostate cancer cells relied on oxidative phosphorylation and shifted toward a reliance on glycolysis for ATP generation during the metastatic stage of cancer. By contrast, the metabolic profile of pluripotent ESCs is featured by high glycolytic activity to compensate cell proliferation, in particular, during hypoxic state of the inner cell mass before implantation of the embryo [105]. However, cancer cells use different metabolic alterations and nutrient adaptation during metastatic stages that lead to tumorigenesis. To date, there is little data supporting a mechanistic link between SOX2 and mitochondria and glucose metabolism directly [106]. It is thus important to identify each master stem cell factor and its role during different embryonic and cancer stages. Accordingly, these studies indicate that reprogramming procedures in cancer cells mostly lead to the generation of malignant cancer-initiating cells that acquire stem cell-like properties. Furthermore, studies of some stem cell reprogramming factors such as SOX2 have revealed new mechanisms by which they enable metastatic progression, cell-lineage plasticity, and therapy resistance. The hallmark of CSCs can be a combination of different characteristics such as phenotypical and metabolic markers that displayed a kind of signature which can be used as new target. These new CSC gene axes and the stemness genes might be representative of the new CSCs markers.

The identification of cellular plasticity and biomarkers are useful approaches for the study of cancer stemness and new oncogenic pathway. Recently, Yang et al. [107] developed a specific genetically encoded biosensor with 72 barcodes (or 128 biosensors) expressed in each cell. In combination with the use of fluorescence, it is now possible to precisely identify the nature of cellular plasticity and discover new cellular pathways, connections, and mechanisms.

Another approach that has been suggested to be an effective novel cancer therapy is exosome-based delivery of cancer-suppressor proteins, microRNA (miR), or targeted drugs [108]. Exosomes are membrane-bound extracellular vesicles that are much smaller than cells and produced in most eukaryotic cells [109,110]. MicroRNA in exosomes affects protein production in the recipient cells [111]. Because exosomes express markers of the cancer cells of origin, clinical applications such as for biomarkers and therapies are realistic.

For example, exosomes containing miR-21 produced an effective downregulation of the *PDCD4* and *RECK* genes in glioma cell lines [112]. Furthermore, patient-derived exosomes loaded with paclitaxel were superior to paclitaxel-loaded liposomes as a cancer immunotherapy in lung cancer cell lines [113], and exosomes carrying inhibitors of the self-renewal, differentiation, and tumorigenesis-related genes (e.g., miRNAs or siRNAs for *TGFβ*, *Wnt*, *Hippo*, etc.) of CSCs are possible therapies for cancer treatment [108]. Taken together, these data suggest that clinical application of exosome-based therapies for various cancers should be addressed in the near future.

## 8. Elusive Problems Faced to Eliminate CSCs

In efforts to remove CSCs effectively, a series of existing problems need to be addressed. (i) As mentioned, because the cancer signaling of CSCs is not specific and shares some pathways with normal stem cells, not all the regulatory factors with or without mutations that contribute to CSCs are appropriate for use as therapeutic targets. (ii) The precise features of many CSCs in specific types of cancers are not well characterized [114]. (iii) Since the tumorigenesis activities of CSCs are determined in immune-deficient mice or animals in the absence of an adaptive immune system, they do not recapitulate the biological complexity of cancers in the clinic [115]. (iv) Information about stem cell niches and the microenvironment are not sufficient. Studies of linkages between CSCs and microenvironments should be combined [51]. (v) Novel studies of the signaling and regulatory mechanisms of cancer metabolism, epigenetics, and mitochondrial functions should be explored [116]. (vi) Inhibitors, small molecules, and antibodies, as well as CSC-directed immunotherapy, need to be developed [117]. These problems should be addressed to promote the promising application of biomarkers for CSC niches for cancer therapy.

## 9. Conclusions

Characterization of biomarkers in stem cells is critical for the identification of the biochemical and genetic events that trigger cancer development (Figure 1). Thus, for each biomarker, such as metabolic reprogramming factors, the links with specific cancer commitment and the associated signaling should be clarified to better understand the conversion of normal stem cells to CSCs. Furthermore, these biomarkers will be useful in the future application of therapeutic tools to treat human cancers. In the future, the identification of biomarkers for stem cell markers and the impact of alterations in signaling on these biomarkers during cancer commitment will be required to reveal the cellular mechanisms and key events in cancer induction that might be potential pharmacological targets.

## Figures and Tables

**Figure 1 jpm-12-00715-f001:**
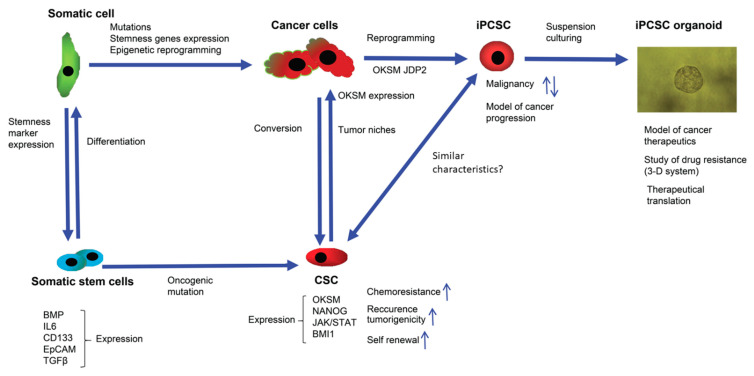
Cancer initiation and the process of cell stemness induction (reprogramming) employs overlapping molecular signaling and epigenetic pathways. Expression of transcription factors—such as OCT4, KLF4, SOX2, NANOG, and JDP2—is necessary for cancer initiation, along with genetic mutations and epigenetic changes. In somatic stem cells, for example MSCs, an increased expression of CD133 by the JAK-STAT 3 pathway activates the migration of MSCs to cancer cells and can increase the number of CSCs. CSC niche factors that induce self-renewal of CSCs stimulate angiogenesis and recruit other cells producing additional factors related to tumor metastasis. Patient-derived iPCSC organoid models are useful for examining the mechanisms of drug resistance and cancer progression.

**Table 1 jpm-12-00715-t001:** Representative biomarkers in each cancer cells. We have modified to add the items based on the following references by Zhao et al. [69], Walcher et al. [44], and ours [1,16].

Cancer Types.	Markers of CSCs
Breast CSCs	CD44/CD24^−^
Breast Carcinoma	ALDH1
Breast Cancer subtype	CD1333, HER2
Prostate CSCs	CD44
Lung CSCs	CD133, ALDH1, CD44
Epithelial CSCs	ALDH1
Glioblastoma	SSEA-1, EGFR, CD44, ID1
Pancreatic CSCs	CD1333, CXCR4, SSEA-1, CD44
Liver metastatic colorectal cancer	EpCAM, CD44, CD24 CEA-CAM, CDX1
Leukemia	CD34, CD38^−^
Gastric CSCs	HER2, APC, p53, kRAS, PTEN, LGR5, CCKBR, RHOA, CDH-1, SMAD5, ATP4B, PGA3

## Data Availability

Please contact the corresponding author for such requests.

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
