# Peer review of "Biomarkers of Cancer Stem Cells for Experimental Research and Clinical Application"

_jpm, 2022, doi:10.3390/jpm12050715_

Round 1

Reviewer 1 Report

Biomarker Identification and Application of Experimental and Therapeutic Cancer Stem Cell Research

To our knowledge, there is a general consensus that some important cancers possess cancer stem cells which have the capacity of self-renewal, pluripotency, and plasticity which leads to therapeutic escape, metastasis and tumour relapse. Genetic, epigenetic, metabolism, and microenvironment influences. CSCs are characterized by surface markers (CD44, CD133, CD34, EpCA,…), by stemness transcription factors OCT3/4, SOX2, KLF4, C-MYC, NANOG, and are associated with pathways (Wnt/TCF, STAT3, NF-kB..). CSCs can be characterized at the transcriptional, posttranscriptional, epigenetic, metabolic and micro environmental level. At present, predominantly surface biomarkers are targeted by immunotherapy. All this you can also find in the review entitled:

“Biomarker Identification and Application of Experimental and Therapeutic Cancer Stem Cell Research”

The authors' review is divided into 8 sections and therefore clearly arranged. With 100 citations, the bibliography is relatively short but up to date with publications from recent years.

Question: is “Experimental and Therapeutic Cancer Stem Cell Research” a phrase or term? Otherwise shouldn’t it be better to reorder the title starting with “Biomarkers of Cancer Stem Cells…”?

There are some concerns:

In the abstract second sentence: to our understanding studies on the heterogeneity of tumours led to the identification of CSCs, and therapy accompanying observations led to deeper insight. What is meant by mechanistic investigation?

Introduction: the last sentence is incomprehensible, better to refer to 1.) neoplasia development and 2. heterogeneity in two separate sentences.

Section 2: iPSCs are introduced without reference to title and abstract, mentioning the iPSC in advance would be helpful and would make the article flow more smoothly. Furthermore, the iPSC section follows an introduction of tumour genesis, looks like an inset and disrupts the flow of reading.

Section 3 second paragraph: long sentence should be subdivided for better understanding.

Figure 1: the designation of the figure (Fig. 1) appears repeatedly.

Comment:

Most of the themes were well developed, but nevertheless, the manuscript suffers from clear conceptual weaknesses. For instance, section 3, here the reports are reviewed in a very detailed way (much more than in the whole text body), however, this special emphasis by the authors is not reflected in the following conclusion, instead an absolute general statement is given without relation to the experimental results.

Conclusions in sections 4 and 5 are equally general and not very specific. Like before, the authors do not develop a realistic association to therapeutic approaches or perspectives.

Furthermore, we found a contradiction: while the so called stemness transcription factors (OCT4, ..) are necessary for cancer development (in text block but also in the explanation of figure 1), the authors state in section 7, that the role of these factors in cancer commitment remains unknown.

Summarized a thorough revision of the manuscript is necessary. The body of the text needs a thorough revision to enable better comprehensibility, long nested sentences should be avoided. In our opinion the authors should especially focus on the conclusions, which could be drawn by each section showing that the citations were selected carefully.

Author Response

Dear Editor:

We are pleased resubmitted our revised review article by S. Saito et al. entitled “Biomarkers of cancer stem cells for basic and clinical application (Old title; Biomarker identification and application of experimental and therapeutic cancer stem cell research) ”. We have revised the text to take all criticism and suggestions raised by reviewers I and II and add the description to reply for understanding the text well. We have added one subsection “Elusive problems faced to eliminate CSCs” in the text to remove cancer development and revised the text to understand the conceptual flow and the aim of this review article.

       We hope these revisions will be satisfactory to the reviewers.

Each point raised by the reviewer are answered as following.

[Reviewer 1]

  1. The authors' review is divided into 8 sections and therefore clearly arranged. With 100 citations, the bibliography is relatively short but up to date with publications from recent years. Question: is “Experimental and Therapeutic Cancer Stem Cell Research” a phrase or term? Otherwise shouldn’t it be better to reorder the title starting with “Biomarkers of Cancer Stem Cells…”?

(Reply) Thank you very much for your kind suggestion. As suggested by the reviewer, we added the recent references of the text to increase the references and add the recent references. Regarding the title, as suggested by the reviewer, we have changed the title of our text to “Biomarkers of cancer stem cells for basic and clinical application”.

  1. There are some concerns: In the abstract second sentence: to our understanding studies on the heterogeneity of tumours led to the identification of CSCs, and therapy accompanying observations led to deeper insight. What is meant by mechanistic investigation?

(Reply) As suggested by the reviewer, we have added the word as the “cancer development” as the mechanistic investigation in the “Abstract” section to let the author understood well. We also modified the sentences in Abstract section.

  1. Introduction: the last sentence is incomprehensible, better to refer to 1.) neoplasia development and 2. heterogeneity in two separate sentences.

(Reply) We have separated it into two sentences to let the audience understood the contents easily.

Section 2: iPSCs are introduced without reference to title and abstract, mentioning the iPSC in advance would be helpful and would make the article flow more smoothly. Furthermore, the iPSC section follows an introduction of tumour genesis, looks like an inset and disrupts the flow of reading.

(Reply) As suggested by the reviewer, we have moved the part of iPSCs explanation before the tumorigenesis section and added the references.

Section 3 second paragraph: long sentence should be subdivided for better understanding.

(Reply) As suggested by the reviewer, we have subdivided the long sentence to two sentences.

Figure 1: the designation of the figure (Fig. 1) appears repeatedly.

(Reply) As suggested by the reviewer, we have deleted the letters in Table 1 and Figure 1.

Comment:

Most of the themes were well developed, but nevertheless, the manuscript suffers from clear conceptual weaknesses. For instance, section 3, here the reports are reviewed in a very detailed way (much more than in the whole text body), however, this special emphasis by the authors is not reflected in the following conclusion, instead an absolute general statement is given without relation to the experimental results.

(Reply) As suggested by the reviewer, we have developed the conceptual possibilities to explain the contradict output of iPCCs. Although we have not known the exact reasons of this contradiction yet, we suggest the possibilities of the mutation of driver genes to explain this these controversial issues and described in section 3. We also show the possible mechanism of the mutation of CSCs marker genes by driver mutations and passenger mutations and the effect of the niches for thinking of the identification of the biomarkers in the latter part of section 5. Indeed, we knew that the mutation of the driver like TP53 is critical for the tumor developments and the tumor suppressions. This possible explanation was added in the middle part of sections 7.

Conclusions in sections 4 and 5 are equally general and not very specific. Like before, the authors do not develop a realistic association to therapeutic approaches or perspectives.

(Reply) As suggested by the reviewer, we have added the explained the importance of the stemness genes and driver and passenger mutations to develop the cancer. The parts of the association with therapeutic approaches and perspectives are described in section 7 and new section 8 in the revised text.  

Furthermore, we found a contradiction: while the so called stemness transcription factors (OCT4, ..) are necessary for cancer development (in text block but also in the explanation of figure 1), the authors state in section 7, that the role of these factors in cancer commitment remains unknown.

(Reply)

Although we nee the further investigation, the function of OCT4 in reprogramming of normal MRFs and cancer cell lines might be different, or the driver mutation of TP53 or p16Ink4a methylation might be different. We describe these possibilities the section 7.

Summarized a thorough revision of the manuscript is necessary. The body of the text needs a thorough revision to enable better comprehensibility, long nested sentences should be avoided. In our opinion the authors should especially focus on the conclusions, which could be drawn by each section showing that the citations were selected carefully.

(Reply) As the reviewer suggested us, we have basically changed the section contents to solute the possible problem in the biomarkers which are established in stemness genes and driver/passenger mutations and the biomarkers. In addition, the possible problems faced to exclude the CSCs are presented in section 8.

Reviewer 2 Report

This is a comprehensive review of the current knowledge about cancer stem cells markers and their potential merits as therapeutic targets. The review is well structured and addresses one of the hottest topics in contemporary  cancer research.

Some minor comments: 

i) Page 3 : “induced pluripotent cancer cells (iPCCs)” mentioned twice

ii) Section 3: There seems to be a logical contradiction in the statement that cell reprogramming by using cancer-initiating genes while being a controversial methodology can contribute to developing new therapies for targeting CSCs. Especially considering existing controversies regarding the effects of iPCCs reprogramming on the tumorigenic capacity. These controversies need to be addressed in more detail.

iii) Section 4: The title “Markers of targets for therapeutics in CSCs” is semantically equivocal. Perhaps “Putative cellular targets” would better reflect the content of this section. Similar suggestion for the title of section 6: “Microenvironmental factors and interplay with CSCs signaling”

iv) Figure 1: Bidirectional arrow connecting somatic stem cells and CSCs in Figure 1 is confusing as it implies a reversal transition from CSCs to somatic stem cells. If the authors imply the effects of CSCs on somatic stem cells, this aspect needs to be explicitly addressed in the text and explained in the legend.

Author Response

Dear Editor:

We are pleased resubmitted our revised review article by S. Saito et al. entitled “Biomarkers of cancer stem cells for basic and clinical application (Old title; Biomarker identification and application of experimental and therapeutic cancer stem cell research) ”. We have revised the text to take all criticism and suggestions raised by reviewers I and II and add the description to reply for understanding the text well. We have added one subsection “Elusive problems faced to eliminate CSCs” in the text to remove cancer development and revised the text to understand the conceptual flow and the aim of this review article.

       We hope these revisions will be satisfactory to the reviewers.

Each point raised by the reviewer are answered as following.

Reviewer 2

This is a comprehensive review of the current knowledge about cancer stem cells markers and their potential merits as therapeutic targets. The review is well structured and addresses one of the hottest topics in contemporary  cancer research.

Some minor comments: 

  1. Page 3 : “induced pluripotent cancer cells (iPCCs)” mentioned twice

(Reply) As suggested by the reviewer, we have revised them.

  1. Section 3: There seems to be a logical contradiction in the statement that cell reprogramming by using cancer-initiating genes while being a controversial methodology can contribute to developing new therapies for targeting CSCs. Especially considering existing controversies regarding the effects of iPCCs reprogramming on the tumorigenic capacity. These controversies need to be addressed in more detail.

(Reply) Thank you very much for your important criticism. We understand what you pointed out.  Thus, we have demonstrated the possible usage of the initial mutation in the Knudson’s hypothesis which postulated the recessive nature of tumor-initiating gene mutations and the mode of inheritance in familial cancer, to examine the TP53 mutation and p16Ink4a methylation and then avoid the second stage mutations as driver mutation. This issue is presented in the sections 5 and 7.

  • Section 4: The title “Markers of targets for therapeutics in CSCs” is semantically equivocal. Perhaps “Putative cellular targets” would better reflect the content of this section. Similar suggestion for the title of section 6: “Microenvironmental factors and interplay with CSCs signaling”

(Reply) AS suggested by the reviewer, we have changed these titles.

  1. iv) Figure 1: Bidirectional arrow connecting somatic stem cells and CSCs in Figure 1 is confusing as it implies a reversal transition from CSCs to somatic stem cells. If the authors imply the effects of CSCs on somatic stem cells, this aspect needs to be explicitly addressed in the text and explained in the legend.

(Reply) as Suggested by the reviewer, we have changed to one direction: from somatic stem cells to CSCs.

The contents of this manuscript are not now under consideration for publication elsewhere. All authors have directly participated in the planning, execution, or data analysis of the study. All authors of this paper have read and approved the final version submitted and co-authors of this paper have read and approved the final version submitted and co-authoring in this manuscript.

Thank you very much for your consideration of our manuscript.

Round 2

Reviewer 1 Report

The manuscript has gained a lot through the revisions, but I would like to make a few remarks, also in the sense of a representative work.

The title has been changed but is 'basic application' perhaps too imprecise, possible is experimental?

Common gene and protein nomenclature should be taken into account throughout the complete text: gene symbols are itacilized, all letters in upper case, full gene names are not itacilized, no upper cases; protein symbols are not itacilized; Gene symbols from mouse: first letter upper case, iticalized.

The new appendix to chapter 5 (Cell Surface) deals with genetic variations that are not to be classified under the chapter heading. Better omitted, as the section also contains few facts.

On page 7, the authors claim to have found out something about TP53, but the corresponding citations do obviously not belong to the authors (99-101).

The middle text section on page 8 (beginning with "Therfore, the combination...") contains a ring closure: "..new targets [.]should be clarified to establish new targets...".  I assume the authors mean that the hallmark of CSC can be a combination of different charcteristics (phenotypical and metabolistc markers) that display a kind of signature which can be used as (new) target.

Author Response

Editor-in-Chief

Journal of Personalized Medicine

Editorial office

MDPT, St. Alban-Anlage 88, 4052 Basel

Switzerland

Journal: Journal of Personalized Medicine
Special Issue: Biomarker Identification and Application of Cancer Stem Cells

Manuscript ID: jpm-1643566R2

April 22, 2022

Dear Editor:

We are pleased resubmitted our revised review article by S. Saito et al. entitled “Biomarkers of cancer stem cells for experimental research and clinical application (Old title; Biomarker identification and application of experimental and therapeutic cancer stem cell research) ”. We have revised the text to take all criticism and suggestions raised by reviewers I.

We hope these revisions will be satisfactory to the reviewers.

Each point raised by the reviewer are answered as following.

==
The manuscript has gained a lot through the revisions, but I would like to make a few remarks, also in the sense of a representative work.

The title has been changed but is 'basic application' perhaps too imprecise, possible is experimental?

(Reply) As suggested by the reviewer, we have changed the title to use the word of “experimental”; Biomarkers of Cancer Stem Cells for Experimental Research and Clinical Application”.

Common gene and protein nomenclature should be taken into account throughout the complete text: gene symbols are itacilized, all letters in upper case, full gene names are not itacilized, no upper cases; protein symbols are not itacilized; Gene symbols from mouse: first letter upper case, iticalized.

(Reply) As suggested by the reviewer, we corrected the signatures of genes and proteins as well as human and mouse items.

The new appendix to chapter 5 (Cell Surface) deals with genetic variations that are not to be classified under the chapter heading. Better omitted, as the section also contains few facts.

(Reply) Thank you for the kind suggestion, we have changed the title of the chapter 5 which are included genetic variations. We think this chapter 5 is critical for the introduction of the ordinal cancer markers. Thus, the tri-relationship of Stemness genes, Driver mutations and Cell surface/genetic markers should be combined and reconsidered as the novel markers of cancer stemness genes. We believe that they are representative as the new biomarkers for the cancer developments. Thus, we included this chapter 5. Previously the reviewer claimed the number of the references. Thus, this time it might be better to include this chapter.

On page 7, the authors claim to have found out something about TP53, but the corresponding citations do obviously not belong to the authors (99-101).

(Reply) As criticized by the reviewer we have corrected the exact citation of TP53 in cancer reprogramming by ours (references 1, 41, 42, 45) and the TP53 in normal reprogramming (references 99-101).

The middle text section on page 8 (beginning with "Therfore, the combination...") contains a ring closure: "..new targets [.]should be clarified to establish new targets...".  I assume the authors mean that the hallmark of CSC can be a combination of different charcteristics (phenotypical and metabolistc markers) that display a kind of signature which can be used as (new) target.

(Reply) As suggested by the reviewer, we have revised these sentences in this text.

The contents of this manuscript are not now under consideration for publication elsewhere. All authors have directly participated in the planning, execution, or data analysis of the study. All authors of this paper have read and approved the final version submitted and co-authors of this paper have read and approved the final version submitted and co-authoring in this manuscript.

Thank you very much for your consideration of our manuscript.

Sincerely yours,

Kazunari K(Kazushige). Yokoyama, Ph.D.

Graduate Institute of Medicine

Kaohsiung Medical University

100 Shih-Chuan 1st Road, San Ming District

807 Kaohsiung, Taiwan

Phone:+886-7-312-1101, ext. 2729

Fax:+886-7-313-3849
